# Exploring the Molecular Players behind the Potentiation of Chemotherapy Effects by Durvalumab in Lung Adenocarcinoma Cell Lines

**DOI:** 10.3390/pharmaceutics15051485

**Published:** 2023-05-12

**Authors:** Marika Saar, Jana Jaal, Alvin Meltsov, Tõnis Laasfeld, Helen Lust, Sergo Kasvandik, Darja Lavogina

**Affiliations:** 1Institute of Clinical Medicine, Faculty of Medicine, University of Tartu, 50406 Tartu, Estonia; 2Institute of Pharmacy, University of Tartu, 50411 Tartu, Estonia; 3Pharmacy, Tartu University Hospital, 50406 Tartu, Estonia; 4Haematology and Oncology Clinic, Tartu University Hospital, 50406 Tartu, Estonia; 5Competence Centre on Health Technologies, 50411 Tartu, Estonia; 6Department of Genetics and Cell Biology, GROW School for Oncology and Developmental Biology, Maastricht University, 6200 MD Maastricht, The Netherlands; 7Institute of Chemistry, University of Tartu, 50411 Tartu, Estonia; 8Department of Computer Science, University of Tartu, 51009 Tartu, Estonia; 9Proteomics Core Facility, Institute of Technology, University of Tartu, 50411 Tartu, Estonia

**Keywords:** lung adenocarcinoma, cisplatin, pemetrexed, durvalumab, proteomics, chemosensitivity, resistance

## Abstract

Immune checkpoint inhibitors are increasingly used in combination with chemotherapy for the treatment of non-small cell lung cancer, yet the success of combination therapies is relatively limited. Thus, more detailed insight regarding the tumor molecular markers that may affect the responsiveness of patients to therapy is required. Here, we set out to explore the proteome of two lung adenocarcinoma cell lines (HCC-44 and A549) treated with cisplatin, pemetrexed, durvalumab, and the corresponding mixtures to establish the differences in post-treatment protein expression that can serve as markers of chemosensitivity or resistance. The mass spectrometry study showed that the addition of durvalumab to the treatment mixture resulted in cell line- and chemotherapeutic agent-dependent responses and confirmed the previously reported involvement of DNA repair machinery in the potentiation of the chemotherapy effect. Further validation using immunofluorescence also indicated that the potentiating effect of durvalumab in the case of cisplatin treatment was dependent on the tumor suppressor RB-1 in the PD-L1 weakly positive cells. In addition, we identified aldehyde dehydrogenase ALDH1A3 as the general putative resistance marker. Further studies in patient biopsy samples will be required to confirm the clinical significance of these findings.

## 1. Introduction

Non-small cell lung cancer (NSCLC) is one of the leading and deadliest types of cancer, causing annually 1.8 million deaths worldwide [1]. The treatment of NSCLC depends on the stage, histology, and tumor cell features, including genetic driver alterations and the expression of markers that predict the efficacy of immunotherapy (e.g., programmed death-ligand 1, PD-L1) [2]. Despite the emergence of targeted therapy strategies, chemotherapy remains the backbone of neoadjuvant and adjuvant treatment before and after surgery in early-stage NSCLC as well as the optimal first-line treatment option in metastatic NSCLC [3,4]. In recent decades, however, the choice of therapies has been complemented by PD-1/PD-L1 axis-targeting immune checkpoint inhibitors (ICIs), which are frequently used in combination with chemotherapy.

A published neoadjuvant chemo-immunotherapy trial showed that both event-free survival and the rate of pathological complete response were better in patients who had received a combination of chemotherapy and ICI nivolumab, although the efficacy depended on the histologic type, used chemotherapy, and PD-L1 expression [5]. Moreover, according to the clinical studies in non-squamous NSCLC patients with stage IV disease, combining ICI durvalumab with the usual chemotherapy regime resulted in a significant increase in the median survival as compared to the effect of chemotherapy alone (e.g., 13.3 months vs. 11.7 months according to the POSEIDON study [6]). Still, not all patients benefit from ICI therapy and chemotherapy–ICI combinations, and in some cases, ICI-induced tumor hyperprogression has been reported [7]. The molecular mechanisms behind the different therapeutic responses are still unclear, and while the immune system-mediated aspects have been widely explored by others, we became interested in whether the proteomic profile of the tumor itself might serve as the major factor determining therapy success. As addressing such a hypothesis is hardly possible in models with a functional immune system, we have limited our studies to in vitro settings using tumor cell lines.

Previously, we have reported the results of extensive screening of seven different chemotherapeutics in the presence or absence of four different ICIs in two cell lines, HCC-44 (high expression levels of PD-L1) and A549 (low expression levels of PD-L1) [8]. Among the tested ICIs, durvalumab showed the most pronounced chemotherapy-potentiating effect, yet also highlighted differences between the used cell lines and the chemotherapeutic drugs. We also demonstrated that the potentiating effect of durvalumab was mostly accompanied by increased DNA damage in cells, but the exact molecular mechanisms behind the potentiation remained unexplored.

Here, we set out to study the proteomic profiles of HCC-44 and A549 cell lysates collected following the 48 h treatment of cells with durvalumab (D), cisplatin (C), pemetrexed (P), or the corresponding mixtures (D + C or D + P). During treatment, two populations of cells arise—some cells undergo apoptosis, yet some survive the treatment—and our goal was to identify both the sensitivity and the resistance markers that determine the success of the combination treatment as compared to the treatments with individual chemotherapeutic agents. For this, we carried out label-free mass spectrometry profiling of the proteome, identified targets enriched in various treatment schemes, and validated the observed trends with an alternative approach represented by immunofluorescent studies combined with automated image analysis.

## 2. Materials and Methods

### 2.1. Chemicals, Cell Lines, and Equipment

The human non-small cell lung carcinoma (adenocarcinoma) cell line HCC-44 and the human lung carcinoma (adenocarcinoma) cell line A549 were from the Leibniz Institute DSMZ (German Collection of Microorganisms and Cell Cultures GmbH). The solutions and growth medium components for the cell culture were obtained from the following sources: phosphate-buffered saline (PBS), fetal bovine serum (FBS), L-glutamine, Dulbecco’s Modified Eagle’s medium (DMEM), and Roswell Park Memorial Institute medium (RPMI-1640)—Sigma-Aldrich (Steinheim, Germany); a mixture of penicillin, streptomycin, and amphotericin B—Capricorn (Ebsdorfergrund, Germany). For the treatment of cells, durvalumab (Imfinzi by Astra Zeneca, Södertälje, Sweden), cisplatin (Accord; Utrecht, Netherlands), and pemetrexed (Selleckchem; Munich, Germany) were used.

The cells were grown at 37 °C in a 5% CO_2_ humidified incubator (Sanyo; Osaka, Japan). The number of seeded or collected cells was counted using a TC-10 cell counter (Bio-Rad; Hercules, CA, USA). During the sample treatment, prior to proteomics, the cells were grown on 10-cm clear cell culture-treated Petri dishes (Thermo Scientific™ BioLite™; Rochester, NY, USA). For the microscopy studies, the cells were grown on 96-well tissue culture-treated Ibidi black μ-plates (ibidi GmbH, Gräfelfing, Germany).

For the proteomics experiments, dithiothreitol (DTT) was purchased from VWR Life Science (Radnor, PA, USA), chloroacetamide (CAA), ammonium bicarbonate (ABC), methylamine, sodium dodecyl sulfate (SDS), urea, and thiourea from Sigma Aldrich (St. Louis, MI, USA). All chemicals were of proteomics grade or ≥99% purity. Lys-C and dimethylated trypsin used for protein digestion were purchased from Wako Chemicals (Osaka, Japan) and Sigma Aldrich (St. Louis, MI, USA), respectively. All organic solvents used for the proteomics were of LC/MS grade from Honeywell (Charlotte, NC, USA).

The liquid chromatography with tandem mass spectrometry (LC/MS/MS) apparatus consisted of a Dionex (Sunnyvale, CA, USA) Ultimate 3000 RSLCnano system coupled to a Thermo Fisher Scientific (Waltham, MA, USA) Q Exactive mass spectrometer. The nano-LC setup consisted of a Dionex cartridge pre-column (ID 0.3 mm × L 5 mm, 5 μm C18) and a New Objective (Woburn, MA, USA) emitter column (ID 75 mm × L 50 cm) packed with 3 μm C18 particles (Dr. Maisch, Ammerbuch, Germany).

For the fixation of cells in the immunofluorescence (IF) experiments, methanol was obtained from Honeywell (Riedel-de Haën™, Seelze, Germany). For the preparation of the blocking solution, BSA from Capricorn Scientific (Ebsdorfergrund, Germany) and PBS (supplemented with Ca^2+^, Mg^2+^) from Sigma-Aldrich (Steinheim, Germany) were used. All primary antibodies were obtained from Sigma-Aldrich (Saint Louis, MI, USA): rabbit polyclonal antibody against human aldehyde dehydrogenase 1 family member A3 (anti-ALDH1A3; HPA064749), rabbit polyclonal antibody against human ankyrin repeat domain-containing protein 17 (anti-ANKRD17; HPA063731), and rabbit polyclonal antibody against human retinoblastoma transcriptional corepressor 1 (anti-RB1; SAB5700023). The secondary antibody (goat cross-adsorbed antibody against rabbit IgG (H+L), conjugated with Alexa Fluor^®^ 568) and the nuclear stain 4’,6-diamidino-2-phenylindole (DAPI) were from Invitrogen (Eugene, OR, USA). Fluorescence microscopy with immunostained cells was carried out with a Cytation 5 multi-mode reader using a 20× air objective (0.3225 µm/pixel). For the DAPI, a 365 nm LED and a DAPI filter block were used; for the Alexa Fluor^®^, 568 and 523 nm LEDs and an RFP filter block were used.

### 2.2. Proteomics Sample Preparation

HCC-44 or A549 cells (passage number below 20) were seeded in growth medium (RPMI-1640 or DMEM supplemented with 10% FBS) onto Petri dishes (1:4 dilution from the confluent Petri) and grown overnight as in culture. Next, treatment with the following compounds or mixtures in usual growth medium (10 mL volume per Petri) was started: 0.49 mg/mL durvalumab, 1 μM cisplatin, 1 μM pemetrexed, mixture of cisplatin (1 μM) and durvalumab (0.49 mg/mL), or mixture of pemetrexed (1 μM) and durvalumab (0.49 mg/mL). After 48 h, the spent culture media were collected into centrifuge tubes. The cells were rinsed with PBS, detached from the plates using 0.25% trypsin, resuspended in the culture medium, and then combined with the corresponding spent media aliquots to collect both detached dying cells and the surviving population. Then, 30 μL aliquots of the obtained cell suspension (total volume of 6 mL) were taken for counting the non-disintegrated cells (the results are shown in Appendix A). Next, the cells were pelleted by centrifugation (5 min at 800 rcf), and the pellets were washed twice with PBS. Finally, the PBS was removed, and the dry pellets were frozen and stored at −90 °C until all independent experiments (N = 3) were finished.

After transportation on dry ice to the proteomics facility, the pellets were suspended in 10 volumes of 4% SDS, 100 mM Tris-HCl pH 8.5, and 100 mM DTT lysis buffer. The samples were heated at 95 °C for 5 min, followed by probe sonication (Bandelin, Berlin, Germany; 3× 20 s pulses, 50% intensity). Unlysed material was pelleted by centrifugation at 14,000× *g* for 10 min. For the full proteome analysis, 15 µg of protein was precipitated with acetone. Protein pellets were suspended in 30 µL of 7 M urea, 2 M thiourea, 100 mM ABC, 2 mM methylamine solution, followed by disulfide reduction and cysteine alkylation, with 5 mM DTT and 10 mM CAA for 30 min each at room temperature (rt). The proteins were pre-digested with 1:50 (enzyme to protein) Lys-C for 1 h, diluted 5 times with 100 mM ABC, and further digested with trypsin overnight at rt. Peptides were desalted with in-house-made C18 StageTips [9] and reconstituted in 0.5% trifluoroacetic acid.

### 2.3. Label-Free Proteomics

First, 2 µg of peptides was injected onto a 0.3 × 5 mm trap-column (5 µm C18 particles, Dionex) from where they were eluted to an in-house-packed (3 µm C18 particles, Dr. Maisch) analytical 50 cm × 75 µm emitter column (New Objective). Both columns were operated at 40 °C. The peptides were separated at 250 nL/min with an 8–35% A-to-B 120 min gradient. Eluent B was 80% acetonitrile +0.1% formic acid and eluent A was 0.1% formic acid in water. The eluted peptides were sprayed into a quadrupole–Orbitrap Q Exactive HF (Thermo Fisher Scientific) MS/MS using a nano-electrospray source and a spray voltage of 2.5 kV (liquid junction connection). The MS instrument was operated with a top-12 data-dependent acquisition strategy. One 350–1400 *m*/*z* MS scan (at a resolution setting of 60,000 at 200 *m*/*z*) was followed by an MS/MS (R = 30 000 at 200 *m*/*z*) of the 12 most intense ions using higher-energy collisional dissociation fragmentation (normalized collision energy of 26). The MS and MS/MS ion target and injection time values were 3 × 10^6^ (50 ms) and 1 × 10^5^ (41 ms), respectively. The dynamic exclusion time was limited to 45 s; only charge states +2 to +6 were subjected to MS/MS.

### 2.4. Proteomics Data Analysis

The MS raw data were processed with the MaxQuant (version 1.6.15.0) software package [10]. For the identification and quantification of the raw MS proteome data, the UniProt human reference proteome database was used [11]. The database was downloaded (both the canonical and isoform sequences) on 20 September 2020, and contained a total of 97,094 entries. Methionine oxidation and protein N-terminal acetylation were set as the variable modifications, while cysteine carbamidomethylation was defined as a fixed modification. The tryptic digestion rule (cleavages after lysine and arginine without proline restriction) was used for in silico digestion of the database. Only identifications with at least 1 peptide ≥7 amino acids long (with up to 2 missed cleavages) were accepted, and transfer of the identifications between runs based on the accurate mass and retention time was enabled. Label-free normalization with the MaxQuant LFQ algorithm was also applied. The protein and LFQ ratio count (i.e., the number of quantified peptides for reporting a protein intensity) was set to 1. The peptide–spectrum match and protein false discovery rate were kept below 1% using a target-decoy approach. All other parameters were the default.

Statistical software R v4.2.3 and package DEP [12] were used for downstream analysis of the quantified proteins. During preprocessing, proteins that were identified in less than two out of three replicates in at least one condition were filtered out. To account for missing values, the Bayesian PCA imputation method was applied [13]. Prior to differential analysis, the counts were logarithm-transformed. The R package limma was used for differential analysis of all possible combinations of treatments [14]. Proteins were considered significantly different at FDR < 0.1, and the FDR cut-off for the top hits was <0.05.

The final lists were also analyzed using the STRING database online platform (Version 11.5, containing information on 19,303 human proteins [15]) to identify and visualize the protein networks enriched in different treatments in either cell line. Both up- and downregulated proteins featuring FDR < 0.1 were included in the analysis.

### 2.5. IF and Microscopy

The cells were seeded onto the plate with densities of 3000 and 4000 cells per well (for HCC-44 and A549, respectively) and grown overnight. Next, treatment of the cells with 1 μM cisplatin or pemetrexed in the presence or absence of durvalumab was carried out as described above, except that 200 μL working volume per well was used. The pilot experiment was carried out in non-treated cells grown in the usual medium. At 48 h after treatment, the medium was removed, and the cells were rinsed with PBS and fixed directly on the plate with cold methanol (15 min at −20 °C). Afterwards, the methanol was removed, and the cells were washed twice with PBS.

Next, blocking with 1% BSA in PBS (weight/volume) was performed for 1 h at rt, followed by the staining and wash procedures according to the previously reported protocol [16]. All primary antibodies were used at a 1:500 dilution in 1% BSA/PBS, except anti-RB1, which was used at a 1:150 dilution. The secondary antibody was used at a 1:1000 dilution; for the staining of the nuclei, 300 nM DAPI in PBS was applied. The imaging parameters (LED intensity, signal integration time, and camera gain) were first optimized in the manual imaging mode for each antibody, and the same parameters were then used for this antibody for all treatments in all independent experiments (N = 5). The imaging was performed in the automated mode; 25 images per well were taken, and the DAPI channel was used for autofocusing. The examples of antibody staining from a pilot experiment in non-treated cells and cells treated with single chemotherapeutic agents are provided in Appendix A.

### 2.6. IF Data Analysis

The automated image analysis using the Ilastik model and the modified version of the Membrane Tools module of Aparecium 2.0 software [17,18] was carried out as reported previously [16].

For further analysis of the raw IF data, the total intensity of the antibody signal in the nucleus was plotted by pooling the data for all nuclei identified in the identically treated cells in all the independent experiments (N = 5). The normality of the data distribution in each condition was tested using the D’Agostino–Pearson test, and non-Gaussian distribution was confirmed for most of the tested conditions. The statistical significance of the pairwise comparison of the treatments of interest was carried out using the unpaired two-tailed Mann–Whitney U-test (95% confidence level).

### 2.7. Other Software

For general data analysis, GraphPad Prism 6 (San Diego, CA, USA) and Excel 2016 (Microsoft Office 365; Redmond, WA, USA) were used. The workflow figures were prepared using the BioRender web application [19].

## 3. Results

### 3.1. Proteomics in Lysates of HCC-44 and A549 following the 48 h Treatment

For our experiments, we chose the same lung adenocarcinoma cell lines as those used in our previous study—HCC-44 and A549, featuring strong and weak PD-L1 positivity, respectively [8]. For the treatment mixtures, two chemotherapy agents were applied: cisplatin, the most widely used drug in NSCLC platinum-doublet treatment, and pemetrexed, which has been mostly used as a monotherapy [4]. The concentrations of compounds used for the treatment of cells (1 μM chemotherapeutic agents and 0.49 mg/mL durvalumab) were also defined based on the previously reported dose–response curves to avoid massive cell death in HCC-44, a more chemosensitive cell line. Following the 48 h treatment, we collected both detached and attached cells for each treatment condition in each cell line (>0.5 million cells were collected for each sample, see Appendix A), and subjected the obtained lysates to LC/MS/MS. The experimental workflow is summarized in Figure 1A.

#### 3.1.1. Proteomics Quality Assessment

The total number of proteins identified in at least one sample was 5895 (with a total of 5647 proteins in the HCC-44 samples and 5667 proteins in the A549 samples). Appendix A shows the protein coverage across the number of samples prior to filtering and after filtering out the genes that are expressed in at least two replicates in any treatment. The latter population was chosen for further analysis; the distribution of the protein counts in differently treated samples are shown in Appendix A. The relatively high number of the listed proteins and the similar pattern of the count distributions in the different samples indicate that the treatment conditions were chosen appropriately (i.e., the extent of cellular death following exposure to cytotoxic compounds was not overly high). Although the lysis protocol used resulted in preferential enrichment for cytosolic rather than membranous proteins, we could identify PD-L1 in 7 out of 15 HCC-44 samples, yet in only 1 out of 15 A549 samples—which is consistent with the previously reported characteristics of the cell lines [8]. In addition, we could detect the fragments of the IgG light chain (k chain V-III region B6 and/or k chain C region) in all samples treated with durvalumab.

The correlation plots showing clustering based on the similarity of the proteome profiles obtained for the different treatments are presented in Figure 2A,B. In HCC-44, a more chemosensitive cell line with a higher expression of PD-L1, all replicate identical treatments expectedly clustered together. The proteomes of the cells treated with the mixture of drugs clustered between the proteomes of the individual drugs. In A549, the clustering was less systematic, yet at least two out of three replicate identical treatments clustered together. The proteomes of the cells treated with pemetrexed or pemetrexed-containing mixtures formed a clearly separate cluster as compared to the other treatments.

These data are also supported by the principal component analysis (PCA) plots (Figure 2C,D). In HCC-44, all treatments form separate clusters, whereas the durvalumab-only treatment clearly stands out from the treatments utilizing chemotherapy agents. On the other hand, in A549, the cluster corresponding to the durvalumab-only treatment partially overlaps with the cluster corresponding to the treatment with a mixture of durvalumab and cisplatin. Still, the positioning of other clusters indicates substantial differences in the proteome among the treatments.

#### 3.1.2. Analysis of Cellular Networks

We then carried out pairwise comparisons of the protein profiles in differently treated samples within the same cell line, focusing more specifically on samples treated with a mixture of drugs vs. a single drug (Figure 1B,C and Appendix A). In line with the higher chemosensitivity and PD-L1 expression level, the number of significant comparisons (FDR < 0.1) was overall larger in the HCC-44 than in the A549 cell line. Furthermore, comparisons of the mixture (D+C, D+P) vs. the durvalumab-only treatment (D) yielded generally more hits than comparisons of the mixture vs. the single chemotherapeutic treatment (C or P)—this indicates that the effect of the addition of chemotherapy to the treatment mixture was generally more pronounced than the effect of the durvalumab addition.

A set of Venn diagrams, shown in Figure 3, illustrates the number of common proteins that were up- or downregulated in the case of different treatments in the same cell line, or in the case of similar treatments in two different cell lines. Within the same cell line, the number of commonly altered proteins was generally higher for the comparisons of D + C vs. D and D + P vs. D. In addition, for such comparisons, more commonly altered proteins could be identified across the two cell lines. This indicates that the addition of chemotherapeutic agents to the treatment mixtures resulted in a more conserved pattern of changes than the addition of durvalumab. Details on the commonly upregulated proteins are provided in Appendix A.

The top hits among the individual molecular players featuring highly different expression levels in different treatments within the same cell line are outlined in a set of Volcano plots in Figure 4 (comparisons of D + C vs. C and D + P vs. P) and Appendix A (comparisons of D + C vs. D and D + P vs. D). Overall, the addition of durvalumab to the treatment mixture triggered profound changes in the expression levels of the proteins involved in DNA damage recognition/repair. For instance, in the HCC-44 cells, the topoisomerase II alpha (TOP2A) and deoxyuridine triphosphatase (DUT) levels were reduced in the D + C vs. C treatment, while the poly(ADP-ribose) polymerase 1 (PARP1) levels were elevated in the D + P vs. P treatments. In the A549 cells, the DNA damage recognition and repair factor (XPA) and structural maintenance of chromosomes flexible hinge domain-containing protein 1 (SMCHD1) levels were reduced in the D + C vs. C treatments. On the other hand, the addition of a chemotherapeutic agent to the treatment mixture was generally associated with increased levels of the cell cycle-related proteins (indicating cell cycle arrest) and decreased levels of the histones (indicating nucleosome degradation during apoptosis). In line with the literature [20,21,22], treatment with the pemetrexed-containing mixtures triggered increases in DHFR, and treatment with both the cisplatin- and pemetrexed-containing mixtures triggered increases in TYMS across the cell lines. In A549, treatment with the chemotherapeutic drug-containing mixtures also caused an elevation of FDXR.

To expose the cellular networks involving the proteins significantly up- or downregulated in the different treatments, we proceeded with the analysis using the online platform STRING. The picked proteins of interest involved in the networks are listed in Table 1, and the networks are visualized in Appendix A. Overall, the STRING analysis highlighted similar trends to those outlined above. Among the additionally picked players, an increase in the CD274 (PD-L1) levels was highlighted for the D + P vs. P comparison in the HCC-44 cell line. While increased PD-L1 expression was previously reported for pemetrexed treatment [23], it is a valuable notion that the addition of durvalumab to the treatment mixture further enhances this trend. In addition, a well-known tumor-suppressing retinoblastoma transcriptional corepressor 1 (RB1) [24,25,26] was featured as a sensitivity-ensuring marker in several comparisons across the cell lines. Yet another common finding was an increase in the aldehyde dehydrogenase 1 family member A3 (ALDH1A3) levels following treatment with the mixture vs. an individual drug in the case of the HCC-44 cell line. The latter trend represents a survival strategy, as high expressions of ALDH1A3 and the same enzyme family members ALDH1A1 and ALDH3A1 have been linked to metabolic reprogramming, which ensures increased chemoresistance and improved survival of cells under hypoxic conditions [27,28,29].

### 3.2. IF in Fixed HCC-44 and A549 Cells following 48 h Treatment

To validate the adequacy of the proteomic analysis, we utilized the immunostaining of the proteins of interest in fixed HCC-44 or A549 cells pre-treated with individual drugs or mixtures that had resulted in significant changes in the validated protein abundance according to the mass spectrometry data. The quantification of the protein abundance in the IF assay was carried out using a previously reported automated algorithm that detects cell nuclei according to the DAPI staining pattern and quantifies the intensity of the signal in the secondary antibody channel for each identified nucleus. The number of identified nuclei per single treatment was in the order of hundreds or even thousands (3000–4000 cells were initially seeded per well, yet some of the cells detached during the 48 h treatment with cytotoxic drugs or during the washing procedures carried out as a part of the IF protocol). Thus, the IF assay measured characteristics of the population of the treated cells and was hence more sensitive towards the cells with low contents of the protein of interest as compared to the proteomics where the total protein abundance in a sample was examined. The general workflow of the validation technique is summarized in Figure 5A.

For validation, we chose markers that were found to be significantly differently expressed in several of the performed treatment comparisons: sensitivity marker RB1 and resistance marker ALDH1A3. The reported nuclear localization of these markers [30] was well compatible with the utilized validation method. Among the proteins for which reduced levels were found in the mixture- vs. single-drug-treated cells, ankyrin repeat domain-containing protein 17 (ANKRD17) was chosen for validation. ANKRD17 also features nuclear localization [30], yet relatively little information is available on its function, and it was not among the hits picked by the STRING platform (Table 1). Still, given that ANKRD17 has been shown to induce JAK/STAT signaling pathways in a variety of cancers, and increased levels of ANKRD17 correlate with poorer prognosis in lung cancer patients [31,32,33], we considered it an interesting novel player.

All of the antibodies chosen for the immunostaining of the four validated proteins featured nuclear signals (Appendix A). RB1 showed only nuclear localization, the ANKRD17 signal was more pronounced in the nuclear envelope, and ALDH1A3 showed additionally varying degrees of cytoplasmic localization. The latter was also dependent on the treatment of cells and/or treatment-related changes in the cell cycle, as the population of cells with nuclear localization of ALDH1A3 was increased following the 48 h treatment of cells with cisplatin (Appendix A).

The quantitative data from a single representative experiment with each validated protein and each tested comparison are shown in Figure 5B–G, and the pooled data from five independent experiments are presented in Table 2. According to the total signal intensity in the nucleus averaged over a population of nuclei imaged in five independent experiments, the expected trends regarding changes in the protein content were confirmed in the cases of all six validated comparisons. The RB1 level was increased in the comparison of D + C vs. C in the A549 cell line as well as in the comparisons of D + C vs. D and D + P vs. D in the HCC-44 cell line; the ALDH1A3 level was increased in the comparisons of D + P vs. P and D + P vs. D in the HCC-44 cell line; and the ANKRD17 level was decreased in the D + C vs. C comparison in the A549 cell line. The statistical significance of the signal intensity difference (non-parametric Mann–Whitney U-test, *p* < 0.05) was confirmed for two proteins, ALDH1A3 and RB1, thus covering five out of the six tested comparisons. Given that the same trends regarding the variation in the protein contents of differently treated cells were confirmed for all six chosen comparisons in both assays used (mass spectrometry and IF), and given the characteristic differences in these two methodologies, we consider the validation successful.

## 4. Discussion

The first-line therapy in NSCLC is mainly based on the levels of PD-L1 expression in patient biopsies, and in both histological types (adenocarcinoma and squamous cell carcinoma), the first-line treatment includes chemotherapy and immunotherapy combinations. Still, the patients’ benefit from the ICI therapy as well as chemotherapy–ICI combinations has been somewhat limited, necessitating detailed studies on the pattern of the molecular players that might impact the efficacy of such therapies—taking into consideration not only the immune system component but also the tumor itself. This work represents a continuation of our previous studies [8,34] focusing on the investigation of ICI effects and the effect-mediating molecular mechanisms at the level of tumors.

Previously, we showed that different ICIs can either potentiate or depotentiate the cytotoxicity of chemotherapeutic agents in vitro, depending on the individual agent and the lung adenocarcinoma cell line used. By using γH2AX as a DNA damage marker, we have also shown previously that such potentiation may occur via the augmentation of the chemotherapy-induced DNA damage; however, this mechanism was not universal for all agents or cell lines tested [8]. The results of this study offer a mechanistic explanation of the previously published results. For instance, in the case of the D + C vs. C comparison in HCC-44, we previously reported somewhat unexpected decreases in the γH2AX levels [8]. Based on the proteomics data (Table 1 and Figure 4), this is currently related to the reduced levels of TOP2A, which functions upstream of the cell machinery catalyzing the formation of γH2AX [35,36]. On the other hand, in the case of the D + P vs. P comparison in HCC-44, we previously reported increases in the γH2AX levels, mirroring the elevated levels of cell death in the mixture- vs. only pemetrexed-treated cells [8]. According to the proteomics, although the expression of TOP2A was also reduced for this comparison, there were multiple other changes in the DNA damage recognition/repair machinery that affected the overall outcome (e.g., reduction in breast cancer gene 2, BRCA2 [37] and gain in PARP1 [38]; Table 1). The previously reported slight yet statistically significant increase in the γH2AX levels for the D + C vs. C treatment comparison in the A549 cells [8] can in turn be explained by the reduced levels of the DNA damage repair protein XPA (Figure 4); this agrees well with the reports on the positive correlation between XPA levels and cisplatin chemoresistance in NSCLC lines. In this way, our current study also confirms our previous observations regarding the efficacy of D + C treatment not only in the highly PD-L1-expressing cell line HCC-44 but also in the A549 cell line with low PD-L1 expression [8].

According to the validation assay, increases in RB1 could also contribute to the potentiating effect of durvalumab in the case of the D + C vs. C treatment in the A549 cell line (Table 2 and Figure 5). Within the frame of this study, we did not validate the changes in the expressions of proteins related to the DNA damage recognition/repair, as the expression changes observed in the proteomics were relatively small and, thus, difficult to quantify using the IF assay. However, we are aiming at evaluating these proteins in our future studies using other methods and samples. In essence, the levels of all of the proteins of interest identified in this work should be assessed in patient biopsy samples and correlated to the known treatment responses to confirm the clinical relevance of our current findings.

We are not aware of other studies utilizing a similar experimental approach to explore the cancer-driven (i.e., as compared to the immune system-driven) aspects of ICI sensitivity and resistance. Multiple publications [39,40,41,42,43] have reported proteomic studies on NSCLC patients’ blood serum and plasma (often both before and within the course of the ICI-containing treatment schemes), attempting to correlate the pattern of the measured markers to the observed responsiveness of individual patients to therapy. Another study focused specifically on the blood-circulating endothelial-derived extracellular vesicles (EVs) isolated from NSLC patients’ samples [44]. While levels of EV secretion are elevated in cancer and EVs can be considered a reservoir of the cancer-derived proteins [45], these do not represent the full proteome of cancerous tissue due to the characteristics of the vesicle formation mechanism. Another publication explored the known ICI targets in paraffinized NSCLC tissue blocks [46] in an attempt to establish a quantitative cancer proteotype, yet the number of targets was limited to ten proteins only. The closest study to the work reported here summarized the available information on the genomic biomarkers implicated in the checkpoint blockade outcome [47]. From the investigational markers outlined there (sensitivity markers: ARID1A, PBRM1, SMARCA4, SMARCB1, BAP1, APOBEC, PD-L1, POLE/POLD1; resistance markers: mutated EGFR, Keap1, JAK1/JAK2, MDM2, PTEN, STK11, or Wnt/beta-catenin pathway members), our proteomic short-lists contained only PD-L1 (CD274; see Table 1). The lack of other outlined markers can be explained by the differences in the experimental approaches and data analysis, as the aforementioned publication [47] reviewed multiple studies utilizing various techniques, and our proteomic approach is not suitable for the examination of mutations.

This work has several limitations, including the lack of in vivo studies or studies with clinical samples. Nevertheless, this study was carried out using adenocarcinoma cell lines (with high and low PD-L1 positivity), which imitate a relevant clinical situation. Moreover, the knowledge obtained from the well-designed preclinical studies may facilitate the selection of proper drugs for chemotherapy and ICI combinations in the future, which is of paramount importance in neoadjuvant chemo-immunotherapy trials where maximal anti-tumor effects and, therefore, better pathologic complete response rates are desired. In principle, the experimental approach utilized here can be expanded to explore the efficacy of combination therapies that also involve targeted compounds. From an experimental point of view, the utilized mass spectrometry approach can be biased towards the proteins with higher expression levels and could be expanded with phospho-proteomics or other techniques to provide additional data on the changes in the protein activity of different treatments. Furthermore, within the current study, the effects were measured following a single-time-point treatment with a single concentration of each drug, and the interpretation of the noted trends regarding the sensitivity vs. the resistance markers was carried out based on the information regarding the roles of the proteins of interest in the literature. As a more elegant experimentally verified approach, monitoring the proteome changes in time can be applied, and we expect to address this challenge in our future studies using more advanced mass spectrometry techniques [48,49]. However, the trends identified in the given study can serve as a strong basis for future research, enabling the stratification of NSCLC patients for ICI-containing therapy schemes based on the levels and mutation status of the putative sensitivity or resistance markers.

## Figures and Tables

**Figure 1 pharmaceutics-15-01485-f001:**
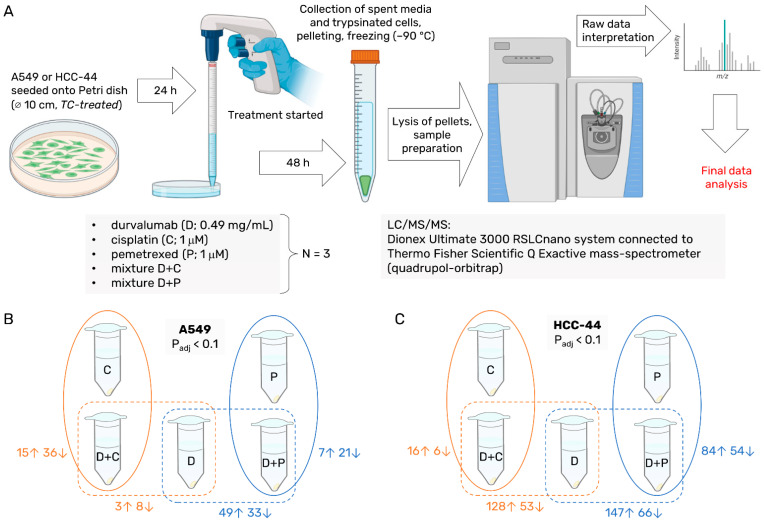
The schematic view of the proteomics experiment and number of hits obtained in this study. (**A**) Workflow: treatment of cells, sample preparation, and mass spectrometry. (**B**,**C**) Number of proteins ((**B**) for A549, (**C**) for HCC-44) identified as significantly enriched in the pairwise comparisons of differently treated cells (FDR < 0.1); ↑ indicates higher abundance in the mixture-treated and ↓ indicates higher abundance in the single agent-treated cells. Abbreviations: C—cisplatin; D—durvalumab; P—pemetrexed.

**Figure 2 pharmaceutics-15-01485-f002:**
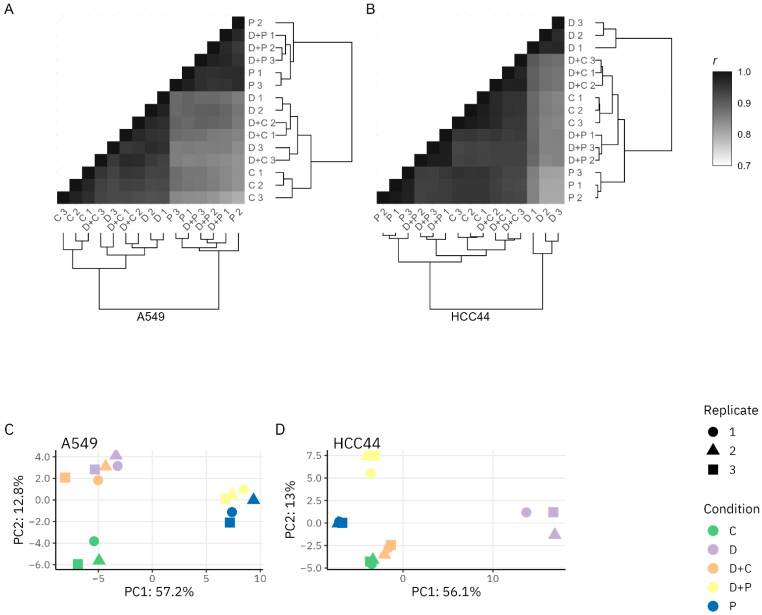
Clustering shows differentiation between the treatments. Correlation with hierarchical clustering (**A**,**B**) and PCA analysis (**C**,**D**) of top variable proteins in A549 and HCC-44 cell lines. Proteins were filtered based on FDR < 0.05 in any compared treatments within a cell line. Correlation analysis was performed with Pearson’s method; the color code is shown on the right. In the case of PCA, different treatments are shown on the right. Abbreviations: C—cisplatin; D—durvalumab; P—pemetrexed.

**Figure 3 pharmaceutics-15-01485-f003:**
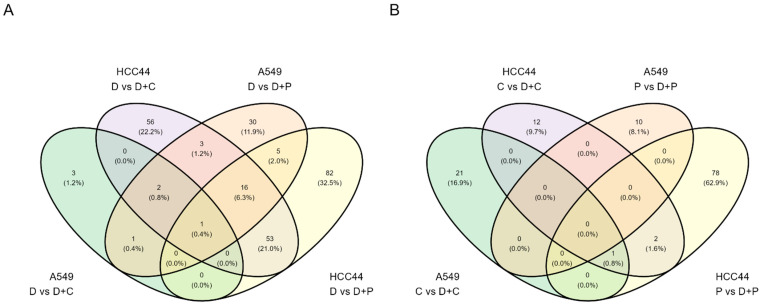
Venn diagrams of the differentially expressed genes in various treatment comparisons in two cell lines. (**A**) Comparison of treatments involving only durvalumab vs. mixture of durvalumab with a chemotherapeutic agent; (**B**) comparison of treatments involving only chemotherapeutic agent vs. mixture of durvalumab with a chemotherapeutic agent. Proteins with FDR < 0.05 in any compared treatments within a cell line were considered significant.

**Figure 4 pharmaceutics-15-01485-f004:**
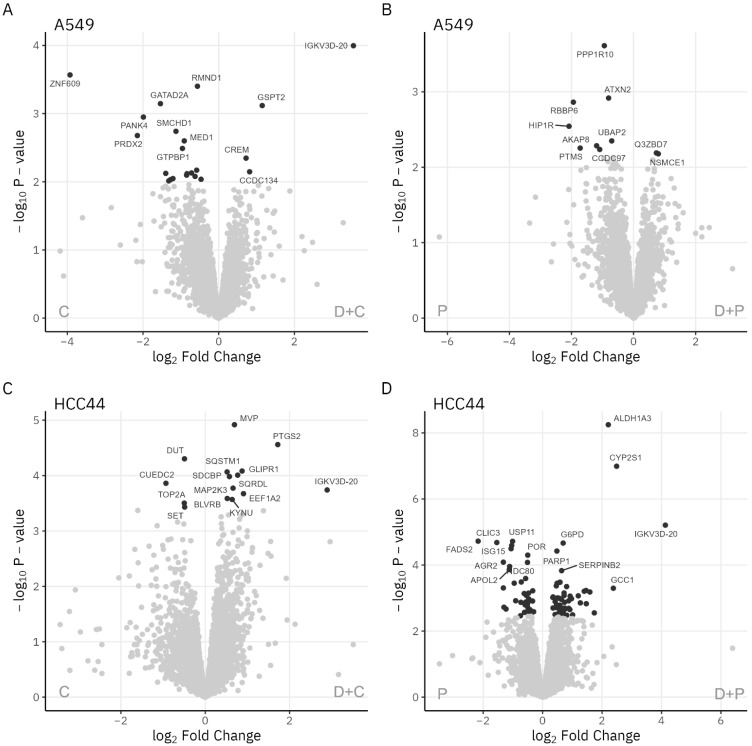
Volcano plots showing effects of chemotherapeutic agent addition to the treatment mixture. (**A**,**B**) A549 cell line, comparison of treatments D + C vs. C (**A**) and D + P vs. P (**B**). (**C**,**D**) HCC-44 cell line, comparison of treatments D + C vs. C (**C**) and D + P vs. P (**D**). Proteins marked in black are considered significantly differentially expressed (FDR < 0.05). IGKV3D-20 is a fragment of durvalumab.

**Figure 5 pharmaceutics-15-01485-f005:**
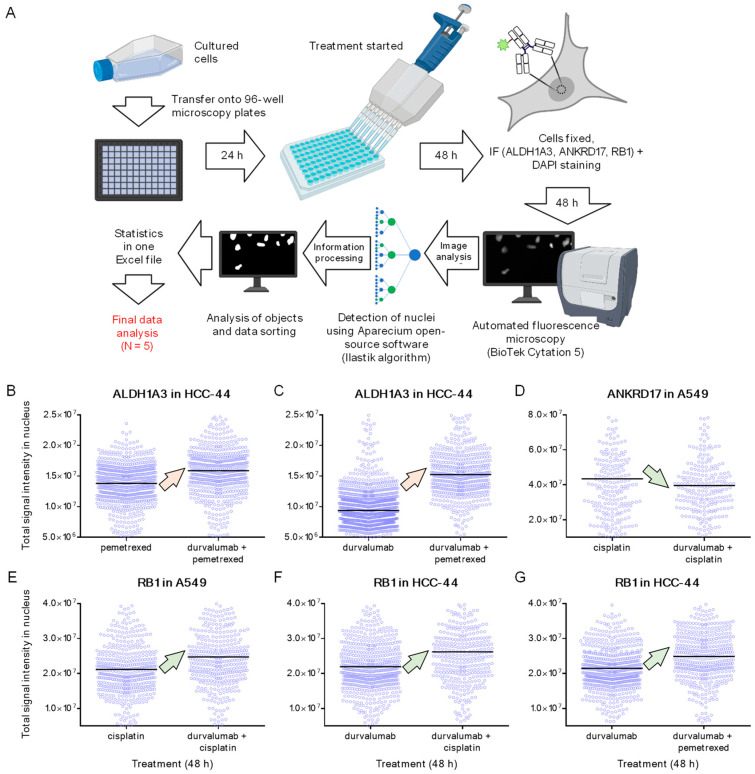
The schematic view of IF experiments. (**A**) Workflow: treatment of cells, sample preparation, fluorescence microscopy, and data analysis. (**B**–**G**) Representative results from a single IF experiment: (**B**,**C**), aldehyde dehydrogenase 1 family member A3 (ALDH1A3) nuclear staining; (**D**) ankyrin repeat domain-containing protein 17 (ANKRD17) nuclear staining; (**E**–**G**) retinoblastoma transcriptional corepressor 1 (RB1) nuclear staining. The compared treatments are shown on the *x*-axis of the graphs. In each graph, each blue circle indicates the signal intensity in an individual nucleus and the thick black line shows the population mean. Arrows show the trend observed in proteomics and IF (up, higher protein levels in the mixture than in the individual drug-treated cells; down, lower protein levels in the mixture than in the individual drug-treated cells). The color of the arrows shows whether the known physiological function of the given protein and the trends observed in this study (increase/decrease in abundance in the mixture-treated cells) indicate participation of validated protein in chemosensitivity (light green) or resistance (light orange) mechanisms.

**Table 1 pharmaceutics-15-01485-t001:** Cellular networks identified by the STRING platform based on the differential abundances of proteins in HCC-44 or A549 samples treated with individual drugs or drug mixtures (proteomics FDR < 0.1).

Cell Line	Compared Treatments ^a^	Protein Hits and Clusters ^b,c^
**HCC-44**	D + C vs. C	DDX58, MAP1LC3B, SQSTM1 DUT, SET, TOP2A EEF1A2, PLEC MAP2K3, PTGS2, SERPINB2
	D + C vs. D	Cell cycle-related cluster (including AURKA, AURKB, BUB1B, CCNB1, KIF11, PLK1, TOP2A, TPX2, TYMS) Cluster-containing DDX58, MAP1LC3B, MAP2K3, PTGS2, **RB1**, SERPINB2, SQSTM1 Cytoskeleton-related cluster (including EHBP1) Histone cluster (including HIST1H4A, HIST1H2AJ, HIST2H3A) RNA-binding protein cluster (including RBM34) Transcriptional activity-related cluster (including SUPT5H)
	D + P vs. P	Cell adhesion and cell–cell communication-related cluster (including CD274, ITGA2, SERPINB2) DNA synthesis and damage response-related cluster (including BRCA2, DHFR, PARP1, PCNA, TOP2A) Growth factor-related cluster (including IGF1R, TGFB2) Metabolic enzyme cluster 1 (including ACAT2, DHCR7, FDPS, MVD) Metabolic enzyme cluster 3 (including **ALDH1A3**, CYP2S1, IDH1, TXNRD1)
	D + P vs. D	Cell cycle-related cluster (including AURKA, AURKB, BUB1B, CCNB1, KIF11, PLK1, TOP2A, TPX2) DNA synthesis-related cluster (including DHFR, TYMS) Histone and transcription regulation cluster (including HIST1H2BC, HIST1H2AJ, **RB1**) Cluster containing MAP1LC3B, MAP1B, SERPINB2, SQSTM1 Cell adhesion and cell–cell communication-related cluster 1 (including CTNNA1, ITGA2, ICAM1, EPHA2) Metabolic enzyme cluster 1 (including **ALDH1A3**, NAMPT, NNMT) Metabolic enzyme cluster 2 (including PGM2L1)
**A549**	D + C vs. C	AKAP1, DPY30 ATP5MF-PTCD1, CHCHD1 CBX4, GATAD2A, **RB1** DTCN5, KIF3A FAM96B, GTPBP1, MLF2 INTS6, ZNF609 MAD1, MED11 RAB5A, RIN1, UVRAG
	D + C vs. D	FDXR, RETSAT HIST1H1B, HIST1H1C, HIST2H3A, KIF4A, MCM2
	D + P vs. P	TWISTNB, WDR55
	D + P vs. D	Cell cycle-related cluster (including CCNB1, KIF11, TOP2A, TPX2) DNA synthesis-related cluster (including DHFR, SHMT2, TYMS) Histone cluster (including HIST1H1B, HIST1H4A)

^a^ C stands for cisplatin, D for durvalumab, P for pemetrexed. ^b^ The abbreviations are arranged alphabetically. ^c^ For sets with multiple players identified, only the largest clusters are listed. Green highlight is used for the rows indicating effect of durvalumab addition to chemotherapeutic agent. The proteins shown in bold were included in the final validation set.

**Table 2 pharmaceutics-15-01485-t002:** Validation results obtained using IF assay (pooled data, N = 5).

Validated Protein	Cell Line and Treatment Conditions ^a^	Trend in IF (Difference in Protein Abundance) ^b^	Statistical Significance of Difference in Protein Abundance ^c^	Relationship to the ICI Treatment Response ^d^
ALDH1A3	HCC-44, P vs. D + P	↑	*p* < 0.001	Increase shows resistance
ALDH1A3	HCC-44, D vs. D + P	↑	*p* < 0.001	Increase shows resistance
ANKRD17	A549, C vs. D + C	↓	ns	Decline shows sensitivity
RB1	A549, C vs. D + C	↑	*p* < 0.001	Increase shows sensitivity
RB1	HCC-44, D vs. D + C	↑	*p* < 0.001	Increase shows sensitivity
RB1	HCC-44, D vs. D + P	↑	*p* < 0.001	Increase shows sensitivity

^a^ The treatment lasted for 48 h; C stands for cisplatin, D for durvalumab, P for pemetrexed. ^b^ Sign ↑ indicates higher abundance in lysates treated with mixture relative to a single-component treatment; sign ↓ indicates lower abundance in lysates treated with mixture relative to a single-component treatment. ^c^ *p*-values calculated using the unpaired two-tailed Mann–Whitney test (95% confidence level); ns, not significant. ^d^ Based on the known functions of the proteins and the abundance changes observed.

## Data Availability

The mass spectrometry proteomics data have been deposited in the ProteomeXchange Consortium via the PRIDE [50] partner repository with the dataset identifier PXD040761. All other data are available from the corresponding author upon reasonable request.

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
