# Peer review of "Exploring the Molecular Players behind the Potentiation of Chemotherapy Effects by Durvalumab in Lung Adenocarcinoma Cell Lines"

_pharmaceutics, 2023, doi:10.3390/pharmaceutics15051485_

Round 1

Reviewer 1 Report

Exploring the molecular mechanisms behind the potentiation of chemotherapy effects by durvalumab in lung adenocarcinoma cell lines.

Marika Saar et al.

In this paper they nicely explore the proteomic differences in two lung adenocarcinoma cell lines (CC-44 and A549) comparing the treatment with cisplatin, pemetrexed and durvalumab in different combinations to observe the different proteomic profile, in order to identify potential markers that explain chemosensitivity or resistance. They also confirmed some of the insights using immunofluorescence. They showed some interesting findings, such as cyclin A2 or RB-1 which explains the potentiation effects in PD-L1 high and low expressed cell lines. They are also able to find ALDH1A3 and ribonuclease 1 to target resistance response. They are also finding some other proteins related that add confidence on the findings.

Minor corrections:

Title: “Exploring the molecular mechanisms behind the potentiation of chemotherapy effects by durvalumab in lung adenocarcinoma cell lines.” 

The title is not reflecting the scope of the paper. In this paper, there is not any exploration of the molecular mechanism behind the biological study: the study explores the proteomic profiling and further validation of some of the findings by immunofluorescence, but not any functional analysis, or any experiment modelling the functionality derived from the proteins found: it is a descriptive methodology, but not any functional or mechanistic experiment. They should adjust the title to the results from the paper. 

The discussion is too long, very convoluted and difficult to follow, needs rewriting. For example this sentence:  “In this study, we aimed at the detailed examination and comparison of the proteome in HCC-44 and A549 cell lines following incubation with 1 mM cisplatin, 1 mM pemetrexed, 0.49 mg/mL durvalumab, or the corresponding mixtures – to gain wider insight regarding the multitude of targets and pathways affected by the chemotherapeutic agents and durvalumab, but also regarding the survival strategies of cells following the 48-h treatment. “

I don’t understand the meaning of: “yielded lists with 181...260 proteins per comparison in a single cell line” and “This yielded lists with 10...76 proteins per comparison in a single cell line”. Is it 10/181 downregulated and 76/260 proteins upregulated? Please clarify.

In the results section, this paragraph is too descriptive, needs quantitative measurements validated by statistical approaches, describing the method you used to calculate them:

Overall, the expected trends regarding changes of protein content dependent on the presence of durvalumab in the treatment solution were confirmed in case of all six validated proteins: ALDH1A3 had higher abundance in HCC-44 cells treated with durvalumab and pemetrexed mixture as compared to pemetrexed alone; ANKRD17 had lower abundance in A549 cells treated with durvalumab and cisplatin mixture (D+C) as compared to cisplatin alone; CCNA2 and INCENP had lower abundance in HCC-44 cells treated with D+C mixture as compared to cisplatin alone; RB1 had higher abundance in A549 cells treated with D+C mixture as compared to cisplatin alone; and RNASEH1 had higher abundance in HCC-44 cells treated with D+C mixture as compared to cisplatin alone. The statistical significance of the signal intensity difference was confirmed for three proteins: ALDH1A3, INCENP, and RB1 (P < 0.05)” 

Major corrections:

It is very stricking that a paper based on mass-spectrometry based proteomics is not showing any quality control for the different proteomic populations analysed: It needs to be improved by showing:

- correlation coefficient plots, such as Pearson or Spearman between replicates, 

- principal component analysis (PCA) showing the clustering from the different types of treatments and cell lines, 

- histograms showing the distribution of the dataset in order to see that all populations are not generally biased by any type of artifact during cell culture or further protein treatment that could affect the quantification,

- Volcano plots to observe the distribution (statistical confidence vs fold change) of the proteins, among other plots that should be shown in order to provide a qualitative visualization of the data.

The statistical method for the mass spectrometry analysis is not optimal for this kind of experiment: RSD introduce uncontrolled biased with only 3 replicates for experiment in the false positive and false negative hints. A more suited statistical analysis should be performed to address this problem: I suggest moderated t-test comparison at 5% p-value and multitest correction (Benjamini-Hochberg) at 5% false discovery rate (FDR) to confirm the positive hints. 

The paper needs improvement in the data treatment and writing to clarify and pint point the novelty. It should not be accepted without addressing those changes. 

Reviewer 2 Report

Authors carried out proteomic analysis of HCC-44 and A549 cells treated with durvalumab in combination with chemotherapeutic agent - cisplatin and pemetrexed to identify resistance markers. 

The manuscript provides good proteomic profiling of these cell lines. I have few suggestions.

Both the cell lines used in the profiling are KRAS mutants. HCC44 is KRAS G12C and A549 is KRAS G12S. Extensive work has been done in last few years to develop pan-KRAS and Mutant specific inhibitors.  G12C inhibitors (Sotorasib) is now clinically approved. Targeted therapies would be preferred choice to treat patients with these mutations. Moreover, KRAS G12Ci and immune checkpoint inhibitors are in clinically trials to enhnaces the efficacy of targeted therapy.

This recent progress decreases the significance of the work provided in manuscript. The authors should discuss how their data can be used predicting combination effects of chemotherapy/targeted therapy with immune checkpoint inhibitors.

I would suggest doing extensive pathway analysis of proteomics data and provides a pictorial view. You can use Metacore, msigDB or enricher. 

Minor comments- 

Please avoid using .... in sentences. for example-

"The first round of sorting (variance cut-off by RSD < 100% and abundance difference cut-off by log2FC >1 or <-1, where positive number indicates higher abundance in mixture-treated samples) yielded lists with 181…260 proteins per comparison in a single cell line (available in Supplementary Table S1)."

Proteomic data submitted to any repository for access?

Reviewer 3 Report

Immune checkpoint inhibitors are becoming widely used for cancer therapy in combination with chemotherapy. But the study of such types of combination therapies is still limited. In this study, the authors treated two lung adenocarcinoma cell lines (HCC-44 and A549) with different combination of inhibitors including cisplatin, pemetrexed, and durvalumab, and analyzed the proteomic response of the cells upon drug treatment via mass spectrometry-based proteome profiling. Their result provides a valuable data source which would be beneficial to clarify the potential effect of certain inhibitors on cancer cell lines. I have several questions/concerns as shown below:

1)      For the treatment of different inhibitors, how did you determine the suitable concentration adopted for the proteomics experiments?

2)      In 2.2 Proteomics sample preparation section, what is the concentration for cisplatin and pemetrexed?

3)      As was described, CAA was used for cysteine alkylation for 30 min at room temperature. Could you help to clarify what does CAA means? And I wonder whether 30 min at room temperature is sufficient for complete alkylation of the cysteine residues.

4)      In the data analysis part, it was described that the UniProt human reference proteome database was used. Could you help to provide the detailed information of the database? Such as how many entries was included and what is the date that the database was downloaded.

5)      For determining the differential proteins, you’d better calculate the p-value between each group of data, so that you will no whether the change is significant or not.

Round 2

Reviewer 1 Report

The manuscritpt posses the quality and the data, supporting the text. It is more clear the scope of the paper and the analysis and discussion are improved. 

Author Response

We thank reviewer once again for the constructive critique that enabled improvement of our manuscript. We will take the principles of the proteomics data analysis suggested in the first review as a guidance for our future studies.

Reviewer 3 Report

The revised version of the article was greatly improved. I have only one further question:

For Figure 4, what's the criteria you adopted for labeling the points in the volcano plots? It seems that some points with ratio between -1 to 1, which means not quite significant, were also labeled. 

Author Response

We thank the reviewer for this clarification request. Indeed, we only used the proteomics FDR cut-off of 0.05 for labelling of points in the Volcano plots, as is indicated in the legends of Figure 4 and Supplementary Figure S7. Our rationale was that in several comparisons that were of most interest for this project (i.e., comparisons of chemotherapeutic reagent-only versus the mixture-treatment, which highlight the role of durvalumab addition), the changes in protein expression levels were relatively moderate. For several proteins for which the difference in expression was found statistically significant based on the FDR cut-off, the binary logarithm values of fold change started from around +/- 0.4, reflecting 1.3-fold change in the protein expression level. However, we decided not to apply the additional cut-off based on the log2FC – on two reasons. First, our choice of treatment conditions (concentrations of compounds and duration of incubation) was based on the rationale to avoid massive amount of cell death – had we chosen higher concentrations and/or prolonged duration, the changes in proteome would have been more dramatic, yet the surviving population of cells would become too limited. Second, most of the identified players with moderate change in the expression levels are enzymes; in essence, 30% increase or decrease in enzyme levels might result in significant changes in the rate of catalysed reaction, which can be mirrored in the physiological outcome. We thus hope that such interpretation of data is acceptable.